# MRI-Based Radiomics Models to Discriminate Hepatocellular Carcinoma and Non-Hepatocellular Carcinoma in LR-M According to LI-RADS Version 2018

**DOI:** 10.3390/diagnostics12051043

**Published:** 2022-04-21

**Authors:** Haiping Zhang, Dajing Guo, Huan Liu, Xiaojing He, Xiaofeng Qiao, Xinjie Liu, Yangyang Liu, Jun Zhou, Zhiming Zhou, Xi Liu, Zheng Fang

**Affiliations:** 1Department of Radiology, The Second Affiliated Hospital of Chongqing Medical University, Chongqing 400010, China; zhanghp@hospital.cqmu.edu.cn (H.Z.); guodaj@hospital.cqmu.edu.cn (D.G.); he_xiaojing@hospital.cqmu.edu.cn (X.H.); qiaoxiaofeng@hospital.cqmu.edu.cn (X.Q.); liuxinjie@hospital.cqmu.edu.cn (X.L.); liuyy@hospital.cqmu.edu.cn (Y.L.); zhoujun@hospital.cqmu.edu.cn (J.Z.); zhouzhiming1127@cqmu.edu.cn (Z.Z.); liuxi@cqmu.edu.cn (X.L.); 2GE Healthcare, Shanghai 201203, China; huan.liu@ge.com

**Keywords:** hepatocellular carcinoma, diagnosis, radiomics, magnetic resonance imaging

## Abstract

Differentiating hepatocellular carcinoma (HCC) from other primary liver malignancies in the Liver Imaging Reporting and Data System (LI-RADS) M (LR-M) tumours noninvasively is critical for patient treatment options, but visual evaluation based on medical images is a very challenging task. This study aimed to evaluate whether magnetic resonance imaging (MRI) models based on radiomics features could further improve the ability to classify LR-M tumour subtypes. A total of 102 liver tumours were defined as LR-M by two radiologists based on LI-RADS and were confirmed to be HCC (*n* = 31) and non-HCC (*n* = 71) by surgery. A radiomics signature was constructed based on reproducible features using the max-relevance and min-redundancy (mRMR) and least absolute shrinkage and selection operator (LASSO) logistic regression algorithms with tenfold cross-validation. Logistic regression modelling was applied to establish different models based on T2-weighted imaging (T2WI), arterial phase (AP), portal vein phase (PVP), and combined models. These models were verified independently in the validation cohort. The area under the curve (AUC) of the models based on T2WI, AP, PVP, T2WI + AP, T2WI + PVP, AP + PVP, and T2WI + AP + PVP were 0.768, 0.838, 0.778, 0.880, 0.818, 0.832, and 0.884, respectively. The combined model based on T2WI + AP + PVP showed the best performance in the training cohort and validation cohort. The discrimination efficiency of each radiomics model was significantly better than that of junior radiologists’ visual assessment (*p* < 0.05; Delong). Therefore, the MRI-based radiomics models had a good ability to discriminate between HCC and non-HCC in LR-M tumours, providing more options to improve the accuracy of LI-RADS classification.

## 1. Introduction

The Liver Imaging Reporting and Data System (LI-RADS) was developed by the American College of Radiology to standardise the interpretation and reporting of imaging for hepatocellular carcinoma (HCC). According to the possibility of liver lesions from definitely benign to definitely HCC, LI-RADS provides 5 categories from LR-1 to LR-5, which play crucial roles in guiding diagnosis and clinical treatment [1]. Previous studies have suggested that the LR-5 class was associated with unfavourable pathological features of resected HCC [2], and confirmed the potential prognostic role of LI-RADS classification, supporting hepatectomy especially for the LR-5 subclass [3]. In the field of liver transplant, although no significant differences were observed between LR-4 and LR-5 HCC probability when LI-RADS was compared with explant pathology [4], every nodule with an intermediate-to-high HCC probability defined by LI-RADS should be given sufficient attention and entered into the Metroticket 2.0 calculator in order to grant appropriate performance [5].

As a special category, LI-RADS M (LR-M) was defined as “probably or definitely malignant but not specific for HCC” in LI-RADS v2017 and as the refined imaging standard in version 2018. Approximately 93% of LR-M lesions based on magnetic resonance imaging (MRI) are pathologically diagnosed as malignancies, including HCC, intrahepatic cholangiocarcinoma (ICCA), combined hepatocellular-cholangiocarcinoma (CHC), and metastases, of which 36% are HCC [6]. Compared with HCC, the therapeutic regimen and prognosis of non-HCC in LR-M are significantly different. For example, ICCA is considered a contraindication to liver transplantation because of its high recurrence rate, and surgical resection is more common. Additionally, controversies persist regarding CHC treatment [7,8]. Although biopsy can clarify the specific pathological types of LR-M lesions, the clinical application of this method is still controversial because of its potential risk of dissemination. Therefore, differentiating HCC from other primary liver malignancies in LR-M noninvasively is critical.

Serum markers are widely used in clinical tumour diagnosis. Compared with those in HCC patients, the carbohydrate antigen19-9 levels were higher in ICCA patients and patients with combined tumours, while the AFP levels were lower in ICCA patients [9]. However, serum markers reveal low specificity and are susceptible to hepatic and extrahepatic conditions. Presently, MRI is widely used in the diagnosis and prognosis of liver tumours. To identify HCC and non-HCC malignancies, ‘not showing delayed central enhancement’ is correlated with HCC, and ‘biliary dilatation or liver surface retraction’ is correlated with ICCA [10]. An intratumoural septum and non-targetoid restriction, as well as an enhancing capsule and blood products in the lesion, may be useful to differentiate HCC assigned to LR-M from non-HCC malignancy on gadoxetic acid-enhanced MRI [11]. Other researchers have found that the T2-weighted imaging (T2WI) targetoid appearance, which is similar to the diffusion-weighted imaging (DWI) targetoid appearance, can be used as a feature of non-HCC malignancies [9]. However, the differentiation efficiency of conventional MRI features in combined tumours and small tumours is limited [12,13]. These MRI features have not yet reached a consensus and are limited by visual evaluation. Therefore, evaluating these differences between HCC and non-HCC patients in LR-M may not be sufficient and objective.

Radiomics, as an emerging field in radiology, provides nonvisual information related to tumour heterogeneity by extracting many quantitative features from high-throughput medical images and quantifying the data. Previous studies have demonstrated the potential of radiomics in differentiating and diagnosing liver tumours, evaluating the therapeutic efficacy, and predicting the prognosis [14,15,16]. At the same time, when comparing different evaluation methods, multisequence-based MR radiomics was significantly more specific than the European Association for the Study of the Liver and LI-RADS criteria for HCC in high-risk patients with hepatitis B virus dominance [17]. However, few studies have been conducted on the performance of radiomics in determining the pathological type of LR-M and differentiating HCC from LR-M.

Therefore, this study aimed to establish and verify the radiomics model by extracting the features of multisequence-based MRI, which was used to discriminate between HCC and non-HCC in LR-M, and the discriminating efficacy was compared with the results of radiologists. Developing a recommended radiomics model for the non-invasive differentiation of HCC from LR-M may guide clinicians in developing appropriate treatment strategies.

## 2. Materials and Methods

### 2.1. Study Population

This retrospective study was approved by the Medical Ethics Committee, and the requirement to obtain written informed consent was waived. To ensure that patients were eligible for this study, two radiologists evaluated liver tumours that confirmed the LR-M MRI diagnostic criteria from picture archiving and communication systems according to LI-RADS v2018 from January 2011 to January 2020. The inclusion criteria were as follows: (1) patients with chronic hepatitis B virus infection via laboratory tests; (2) liver cirrhosis confirmed by pathological examination via liver biopsy or surgery; and (3) pathological results of the tumours obtained through puncture or surgery within one month after MR examination. Patients with liver cirrhosis younger than 18 years, congenital or vascular-related cirrhosis, and who received any treatment for liver tumours were excluded. For patients with multiple lesions, the target lesions matched the surgically resected lesions, and the pathological results were selected for analysis. Any discrepancies in the results were resolved by consensus between the two observers. Finally, a total of 90 patients with 102 tumours were included in this study (Figure 1).

### 2.2. MR Examination

The images of all 90 patients were acquired using a 1.5T MRI system with an eight-channel phased array software coil. Among these 90 patients, 76 were subjected to Scanner 1 (HDxt2012; GE Medical Systems, Fairfield, OH, USA) with the following sequences: (1) axial fat-suppressed (FS) T2WI with a fast spin-echo (FSE) sequence, repetition time (TR)/echo time (TE) = 6316/90.9 msec, slice thickness = 7 mm, interslice gap = 1 mm, field of view (FOV) = 440 × 352 mm and matrix size = 288 × 224; (2) contrast-enhanced FS T1-weighted imaging (T1WI) with a three-dimensional liver acquisition and volume acceleration sequence, TR/TE = 4.2/2.0 ms, slice thickness = 4.8 to 5.4 mm, interslice gap = −1.4 to −2.7 mm, FOV = 420 × 336 mm and matrix size = 320 × 192. The images of another 14 patients were acquired using Scanner 2 (Magnetom Avanto, Siemens, Erlangen, Germany) with the following sequences: (1) axial FS T2WI with an FSE sequence, TR/TE = 2000/112 msec, slice thickness = 7 mm, interslice gap = 1 mm, FOV = 360 × 360 mm, and matrix size = 256 × 256; (2) contrast-enhanced FS T1WI examination with a volume interpolated body examination sequence, TR/TE = 3.9/1.4 msec, slice thickness = 3 mm, interslice gap = 0.6 mm, FOV = 380 × 280 mm and matrix size = 288 × 162. Gadodiamide (Omniscan; GE Health caree, Co., Cork, Ireland) was injected at a rate of 2.0 mL/s in contrast-enhanced scans for a total dose of 0.2 mL/kg body weight. Enhanced arterial phase (AP), portal venous phase (PVP), and delayed phase data were obtained at 22–25 s, 60–65 s, and 150 s, respectively, after contrast injection.

### 2.3. Image Analysis

Image analysis was performed independently by two abdominal radiologists (Y.Y.L. and H.P.Z., with 8 and 12 years of experience in abdominal MRI, respectively), blinded to the patient’s clinical history and pathological diagnosis. Before this analysis, two observers reached a specific agreement for each LR-M feature defined by LI-RADS v2018 and conducted exercises with several cases not included in the study. According to LI-RADS v2018, the LR-M criteria included targetoid or non-targetoid masses. The imaging manifestations of targetoid masses were defined as follows: targetoid dynamic enhancement (including rim arterial phase hyperenhancement (APHE), peripheral “washout,” and delayed central enhancement) and targetoid appearance on DWI or hepatobiliary phase (HBP) (including targetoid restriction and targetoid HBP appearance). Nontargetoid masses were defined as tumours with one or more of the following characteristics: infiltrative appearance, marked diffusion restriction, necrosis, or severe ischaemia, liver surface retraction, and adjacent biliary obstruction. The MR scanning programme of our institution and the DWI and Gd-EOB-DTPA protocols were unnecessary, so the targetoid mass evaluated by the observers mainly included rim APHE, peripheral “washout,” and delayed central enhancement. During the image evaluation, any discrepancies in the results were resolved by the consensus of the two observers.

Visual evaluation ability was tested by another pair of junior and senior abdominal radiologists (X.F.Q. and X.J.H., with 4 and 15 years of experience in abdominal MRI, respectively), who were responsible for marking HCC and non-HCC in selected cases without knowing the pathological diagnosis. Because there are no specific radiological diagnostic criteria to provide reference for the diagnosis of atypical enhanced HCC, two radiologists can only distinguish HCC and non-HCC in LR-M based on their own clinical experience.

### 2.4. Histopathologic Evaluation

All selected lesions obtained histopathological information through surgical resection or puncture, which was confirmed by pathologists with more than 5 years of experience in pathology. For patients with multiple lesions, we used the Couinaud Liver Segmentation method to locate the lesions to ensure that the obtained pathological information matched the lesions.

### 2.5. Image Segmentation and Radiomics Feature Extraction

Radiomics analysis was performed on the axial T2WI, AP, and PVP images. The grey-level standardisation of all the images was executed before the data were downloaded from the picture archiving and communication systems. In-house software (Artificial Intelligence Kit, v3.2.2; GE Healthcare) was used for image registration, and all the voxels were resampled to a uniform pixel size of 1 × 1 × 1 mm^3^. Subsequently, a free open source software package (ITK-SNAP, Version 3.6.0) was used to segment the region of interest (ROI) of the tumour layer-by-layer and automatically merge each layer of ROI into a volume of interest (VOI). The extent of ROIs included the boundary of the tumours as much as possible but did not involve the adjacent background liver tissue. The ROIs of all the tumours were delineated independently by two radiologists (Z.F. and J.Z., with 16 and 9 years of experience in MRI, respectively), and the interobserver reproducibility was evaluated by calculating the intraclass correlation coefficient (ICC).

The radiomics feature extraction process followed the image biomarker standardisation initiative (IBSI) [18]. First, the VOIs of T2WI, AP, and PVP were imported into AK software in batches, and then automatically extracted a total of 851 features in each phase, including the first-order features, shape factors, the grey-level cooccurrence matrix (GLCM), run-length matrix (RLM), grey-level size zone matrix (GLSZM), neighbourhood grey-tone difference matrix (NGTDM), and transform features (including wavelet features). The complete delineation of the workflow is shown in Figure 2.

### 2.6. Radiomics Features Analysis

All included tumours were randomly divided into a training cohort (*n* = 72) and a validation cohort (*n* = 30) according to a 7:3 ratio for modelling and verification. The feature dimensionality reduction and selection were fulfilled as follows. First, the outlier values were replaced by the median value of the particular variance vector once the values were beyond the range of the mean and standard deviation. Standardisation was performed to normalise the data in a specific interval. Second, the max-relevance and min-redundancy algorithm (mRMR) was used to remove the redundant features. Finally, the least absolute shrinkage and selection operator (LASSO) analysis was performed to reduce the redundancy of the features. The regularisation parameter (λ) of LASSO was used to perform 10-fold cross validation and select features with nonzero coefficients (Figure 3). The radiomics score (rad-score) of each tumour was calculated through a linear combination of the valuable features multiplied by their respective coefficients.

### 2.7. Model Construction and Validation

Stepwise logistic regression analysis was used to construct a radiomics model to identify HCC and non-HCC and included T2WI, AP, PVP, and the corresponding combined models. The model was further validated in the validation cohort. Simultaneously, receiver operating characteristic (ROC) curves were generated to evaluate the performance of two radiologists (junior and senior radiologists) and various targetoid masses (rim APHE, peripheral “washout,” and delayed central enhancement) in distinguishing HCC and non-HCC. The discriminative performance of the radiologists, targetoid masses, and different models was compared using the DeLong test.

### 2.8. Statistical Analysis

The ICC was first used to test the consistency of the radiomics features extracted between operators. Cohen’s k statistic was used to assess the consistency of the imaging features and LI-RADS category between two observers. The Kolmogorov–Smirnov test was used to test the normal distribution of continuous variables. A two-sample t-test was performed for normally distributed data, and the Mann–Whitney U test was used for nonnormally distributed data. The identification performance of the radiomics model was quantified by the area under the ROC curve (AUC) and the Delong test. All statistical analyses were performed using R software (version 3.6.0; http://www.Rproject.org). A two-tailed *p* < 0.05 indicated a statistically significant difference.

## 3. Results

### 3.1. Tumour Characteristics

A total of 90 patients (average age, 54.0 ± 11.5 years) with 102 tumours were enrolled in this study and included 71 tumours in the non-HCC group (including 53 ICCAs, 1 CHC, 14 metastases, and 3 carcinosarcomas) and 31 tumours in the HCC group (including 8 well differentiated HCCs, 19 moderately differentiated HCCs, and 4 poorly differentiated HCCs). All tumours were randomly split into training (*n* = 72) and test cohorts (*n* = 30) at a ratio of 7:3, and the positive rates of HCC were 30.6% (22/72) and 30% (9/30) in the training and validation cohorts, respectively. No statistically significant differences were found in the clinical characteristics or laboratory indicators between the groups (*p* > 0.05) (Table 1).

### 3.2. Feature Selection and Radiomics Signature Construction

A total of 851 features were extracted from T2WI, AP, and PVP, and the features with coefficients >0.8 were selected by the ICC test. Finally, 761, 777, and 785 features from T2WI, AP, and PVP were entered into the subsequent analysis. After data standardisation and dimensionality reduction by max-relevance, mRMR, LASSO, and logistic regression analysis, 3 features in T2WI, 7 features in AP, 4 features in PVP, 5 features in T2 + AP, 4 features in T2 + PVP, 3 features in AP + PVP, and 7 features in T2 + AP + PVP were used to establish the radiomics model (Table 2).

### 3.3. Performance of Radiomics Models and Verification

Seven models were established: T2WI, AP, PVP, T2WI + AP, T2WI + PVP, AP + PVP, and T2WI + AP + PVP. The ROC curves of each radiomics model in the training cohort and validation cohort are shown in Figure 4A,B. Among these models, the combined model based on T2WI + AP + PVP showed the best performance. No difference was detected between the training cohort and validation cohort (*p* > 0.05; Delong test) (Table 3). Each tumour’s rad-score in the training cohort and validation cohort is shown in Figure 5.

### 3.4. Classification Performance Verification of Visual Evaluation and Imaging Features

The consistency of the two observers marking rim APHE, peripheral washout, delay central enhancement, and LI-RADS category was substantial, and the Cohen’s kappa coefficients were 0.76, 0.71, 0.74, and 0.81, respectively.

The ROC curves of the junior radiologist, senior radiologist, and each targetoid mass are shown in Figure 6A,B. Delong analysis indicated that the ability of senior physicians to judge HCC and non-HCC in LR-M tumours was significantly higher than that of junior physicians (*p* = 0.007; Delong test). The single and multiple targetoid masses showed moderate discrimination, and no significant differences were found between them (*p* > 0.05 Delong test) (Table 4). The ability of junior radiologists and targetoid masses to distinguish HCC and non-HCC was significantly lower than that of various radiomics models (*p* < 0.05 Delong test), and only the ability of senior radiologists was equivalent to that of radiomics models (*p* > 0.05 Delong test).

## 4. Discussion

In the present study, we developed and validated radiomics models based on MRI to distinguish HCC from LR-M tumours. Our study showed that the radiomics model based on T2WI, AP, and PVP images achieved good results in the training and validation cohorts. All seven models achieved encouraging discrimination performance; among them, the combined model based on T2WI + AP + PVP showed the best discrimination performance. Simultaneously, the identification performance of the radiomics model was better than that of junior radiologists’ visual assessment and pure LR-M tumour imaging feature evaluation.

In contrast to other reporting systems, LR-M, as a special type of LI-RADS, is defined as a malignant tumour other than HCC. However, in actual clinical work, many HCC cases are included in LR-M [19,20], challenging clinicians to formulate treatment strategies. Considering that HCC and non-HCC are significantly different in treatment and prognosis, distinguishing them effectively in LR-M tumours is clinically significant. Relying on the knowledge and experience of radiologists to interpret MRI images is the traditional way to solve this puzzle. In our study, the junior radiologist and senior radiologist showed different abilities in distinguishing the subtypes of LR-M tumours (*p* = 0.007; Delong test). A senior radiologist could better distinguish HCC in LR-M tumours (AUC = 0.799) but still divided 9 of 31 HCCs into a non-HCC group, while a junior radiologist divided 14 HCCs into a non-HCC group. This result is disappointing and will likely lead to a more perplexing follow-up diagnosis and treatment.

Among LR-M tumours, HCC and ICCA account for 36% and 30%, respectively [6,21,22]. CHC, metastases, and sarcomas only account for a small proportion of LR-M tumours. However, these tumours have become a risk factor affecting the visual evaluation diagnosis [23]. Our study also confirmed that distinguishing HCC and non-HCC in LR-M tumours by relying solely on the targetoid mass with visual inspection is challenging. When the tumour had only a single targetoid mass, its ability to discriminate between HCC and non-HCC was general (AUC: 0.685–0.723). Even if the tumour had three targetoid masses concurrently, its AUC to distinguish HCC and non-HCC was also only 0.754. The imaging features of LR-M tumour subtypes overlap, making visual evaluation limited. Additionally, the lack of DWI and HBP images may explain the unsatisfactory results of the present study. Perhaps the dual evaluation strategy of junior and senior radiologists can improve this dilemma, and further research is required.

Radiomics, as an emerging discipline that has emerged in the context of big data, has the characteristics of stable calculation, high reproducibility, and freedom from human subjective initiative interference [14,24]. Regarding liver tumour research, radiomics has provided encouraging results in identifying benign and malignant liver tumours [25], predicting the recurrence of HCC after surgical resection [26] and prognosis after transcatheter arterial chemoembolisation [27], and predicting HCC histological grade [28] and microvascular invasion (MVI) [29,30]. Currently, most radiomics studies on LI-RADS have focused on classification and diagnosis [17,31], and their application in LR-M tumours, particularly the identification of LR-M subtypes, has not yet been reported. Theoretically, although HCC and non-HCC in LR-M have similar imaging manifestations, differences exist in their cell origin, spatial arrangement and distribution of tissue cells, vascular heterogeneity, and other tumour characteristics. These differences cannot be distinguished by visual inspection, but radiomics is a promising approach. In our study, both models based on AP, PVP, and T2WI, or the corresponding combined models, showed good discrimination ability (AUC: 0.768–0.884), which was confirmed in the validation cohort. The discrimination ability of each radiomics model was proved to be significantly better than that of junior radiologists, and was equivalent to senior radiologists, indicating that radiomics can not only provide a better identification method for junior radiologists in the differential diagnosis of HCC and non-HCC in LR-M tumours, but also serve as an important reference method for senior radiologists. Additionally, among the three models of AP, PVP, and T2WI, the AP model was better than the other two models, while a previous study demonstrated that the PVP model showed better performance [27]. This difference may be due to the different research subjects. Their model was mainly established for HCC. However, in the present study, most of the tumours were ICCA, which was only enhanced in the marginal tumour in AP, a finding that was significantly different from the obvious enhancement of typical HCC.

The first-order features describe the distribution of voxel intensities in ROI through the commonly used fundamental matrix but do not involve spatial information. In our study, the first-order feature minimum is the original image-based feature retained by all the other six models except model1 after dimensionality reduction. The original_firstorder_Minimum represents the minimum grayscale intensity of the original image in the ROI region, and the subtle differences between them cannot be recognised by visual evaluation, but the radiomics method can. The radiomics models assembled based on the original_firstorder_Minimum extracted from MR images have been proven to be effective in predicting the efficacy of chemoradiotherapy for advanced cervical cancer and the pathological features of rectal cancer [32,33]. In addition, Skewness (Model4, Model7) and Idmn (Model2, Model4) in the original features were also the key features of participating in model construction. Skewness, as a first-order feature, reflects the asymmetry of image grayscale value relative to the mean value, which can describe the shape of the histogram and the “procrastination” direction of the tail. In the application field of liver tumours, a radiomics model including original_firstorder_Skewness and other features showed good predictive efficiency in predicting the MVI of HCC (AUC:0.858) [34]. Idmn is a second-order texture feature based on GLCM, which describes the inverse difference moment normalised of image grayscale and reflects the homogeneity of image texture; the larger its value is, the smaller the change between different regions of image texture. Huang et al. [35] showed that MRI-derived GLCM_Idmn was among the key features in the predictive models for recurrence-free survival of breast cancer, but the application of this feature in liver tumours needs more studies to confirm. The shape feature Flatness was also found to be an important feature in the construction of Model 7 in our study, which was currently only applied in a few CT-based radiomics models [36]; we hope to verify it in subsequent research based on MR images.

The wavelet transform uses the wavelet function to decompose the original image to obtain wavelet-based features. The features after the wavelet transform often carry more tumour information, which can more accurately reflect the heterogeneity of tumours [37]. In this study, the features obtained after dimensionality reduction were mostly wavelet features. After wavelet transform, the first-order features (Mean, 10Percentile, and Kurtosis) were important constituent features in our model, which were consistent with the features extracted by the previous model for discriminating benign and malignant prostate lesions [38]. The wavelet transformed first-order feature skewness and GLCM feature Idmn were still retained in several models after feature screening, which proved again that the distribution of image grayscale of HCC and non-HCC in LR-M was asymmetric relative to the average value, and the variation and uniformity of image texture in tumour region were inconsistent. The GLCM feature Idn, as another metric feature reflecting the uniformity of the image, standardises the difference by dividing the sum of adjacent intensity values by the total discrete intensity values. A predictive model constructed with wavelet-transformed GLCM_Idn as one of 14 radiomics features proved to be reliable in predicting which patients would develop extrahepatic spread or vascular invasion following initial TACE monotherapy (the AUC of the training cohort and validation cohort were 0.911 and 0.847, respectively) [39]. In addition, in our study, the wavelet transformed first-order feature (RobustMeanAbsoluteDeviation), GLCM features (Imc1, MCC, and DependenceVariance), NGTDM feature (Strength) and GLSZM feature (LargeAreaLowGrayLevelEmphasis) participated in the establishment of some models, indicating that these features probably have great medical value in discriminating HCC and non-HCC in LR-M tumours. However, few studies have explained the correlation between these features and tumour pathophysiology, which is still a very challenging task at this stage. 

Although this study was novel, it also had some limitations. First, this study was a single-centre retrospective study, selection bias was inevitable, and external verification was lacking; further validation is required in multicentre research with larger samples. Second, DWI and HBP were not necessary in our institution’s MR scanning programme, and the corresponding targetoid masses, such as targetoid restriction and targetoid HBP appearance, could not be included in the study. Third, the sample size of this study was small, and the data distribution between the HCC and non-HCC groups was unbalanced. Prospective studies that recruit more patients will help verify and improve the practicality of the model. Finally, HCC in LR-M was not a typical HCC enhancement method, and we could not use the known HCC diagnostic criteria to determine HCC. Therefore, during visual evaluation, two radiologists could only distinguish between HCC and non-HCC according to their own clinical experience, leading to the influence of subjective consciousness on the results of visual evaluation.

## 5. Conclusions

This study provides radiomics models based on AP, PVP, and T2WI for the non-invasive evaluation of HCC and non-HCC in LR-M tumours and verifies that the radiomics methods are superior to junior radiologists’ visual assessment. Thus, more reference methods are provided to classify HCC and non-HCC in LR-M tumours, and a favourable guarantee for junior radiologists is offered to preoperatively diagnose the subtypes of LR-M tumours.

## Figures and Tables

**Figure 1 diagnostics-12-01043-f001:**
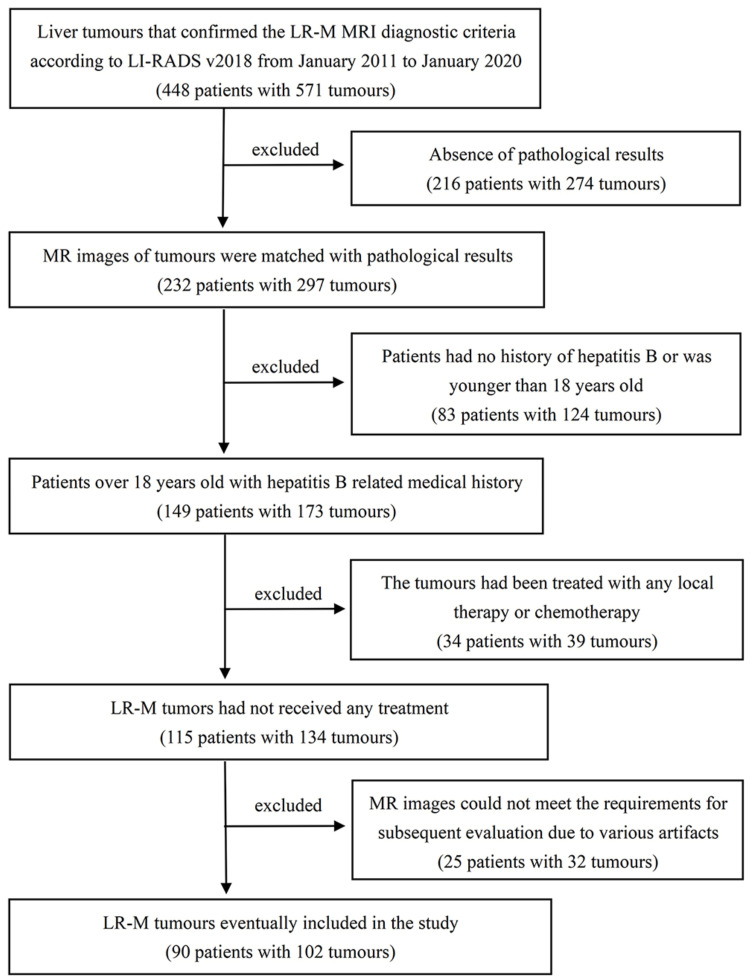
Flowchart showing the inclusion and exclusion of patients.

**Figure 2 diagnostics-12-01043-f002:**
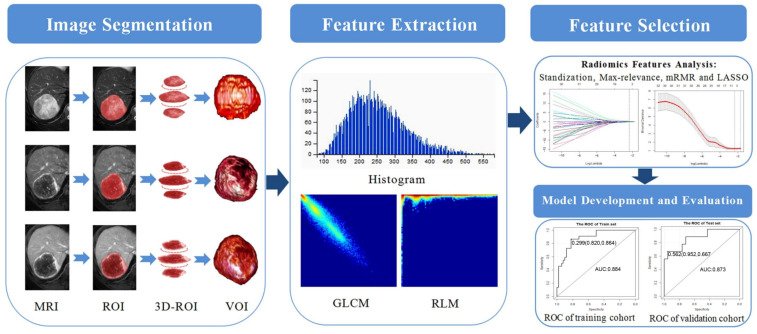
Schematic diagram of the processing and analysis flowchart. ROIs were manually delineated over the whole tumour layer by layer on T2WI, AP, and PVP images and automatically merged into a VOI. Radiomics features were extracted automatically using Artificial Intelligence Kit software. Standardisation, max-relevance, mRMR, and LASSO analyses were used to reduce the redundancy or selection bias of the features. Finally, different models were constructed and verified. MRI, magnetic resonance imaging; ROI, region of interest; VOI, volume of interest; GLCM, grey-level cooccurrence matrix; RLM, run-length matrix; mRMR-40, max-relevance and min-redundancy; LASSO, least absolute shrinkage and selection operator.

**Figure 3 diagnostics-12-01043-f003:**
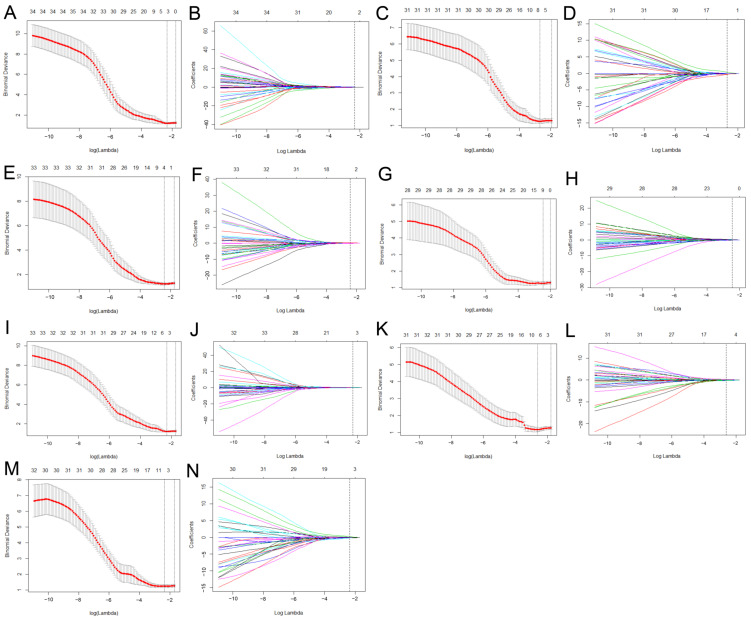
Feature selection using LASSO logistic regression of T2WI (**A**,**B**), AP (**C**,**D**), PVP (**E**,**F**), T2WI + AP (**G**,**H**), T2WI + PVP (**I**,**J**), AP + PVP (**K**,**L**), and T2WI + AP + PVP (**M**,**N**) imaging. The LASSO binary logistic regression model was used to select features and the regularisation parameter (λ) of the LASSO was used to perform 10-fold cross-validation (**A**,**C**,**E**,**G**,**I**,**K**,**M**). The coefficients were plotted against the log (λ) sequence, and the vertical line was drawn at the value selected using 10-fold cross-validation in the ln(lamda) sequence, nonzero coefficients were selected finally (**B**,**D**,**F**,**H**,**J**,**L**,**N**). LASSO, least absolute shrinkage and selection operator.

**Figure 4 diagnostics-12-01043-f004:**
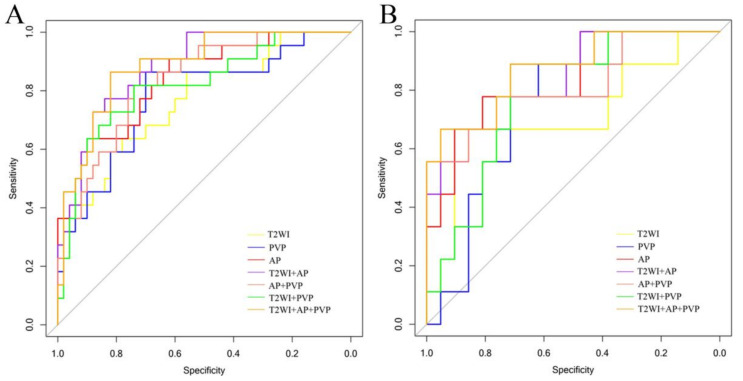
Comparison of the receiver operating characteristic curves of each radiomics model in the training (**A**) and validation (**B**) cohorts.

**Figure 5 diagnostics-12-01043-f005:**
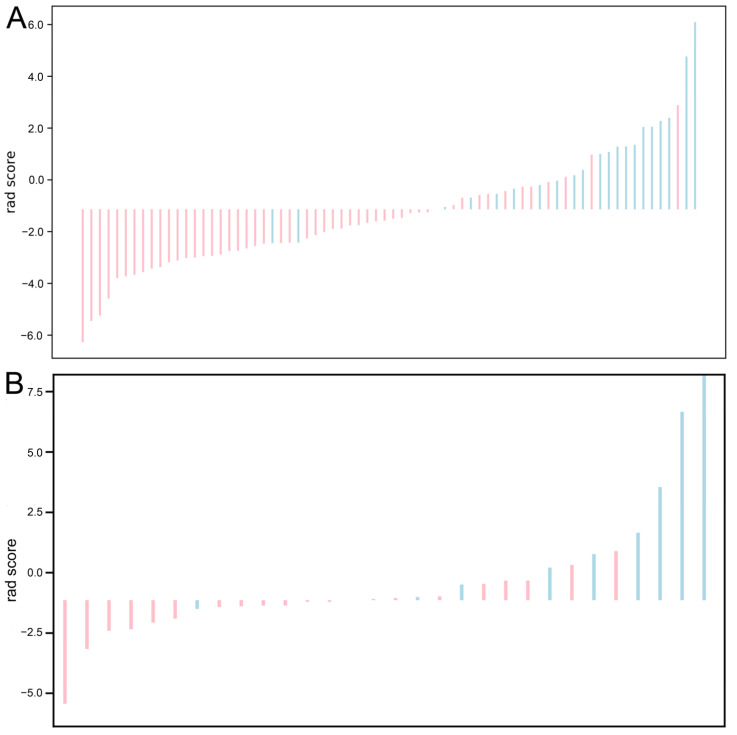
Radiomics score (rad-score) for each tumour in the training cohort (**A**) and validation cohort (**B**). The red bars show the rad-scores of the tumours with significant non-HCC, and the blue bars show the rad-scores of those with significant HCC. HCC, hepatocellular carcinoma.

**Figure 6 diagnostics-12-01043-f006:**
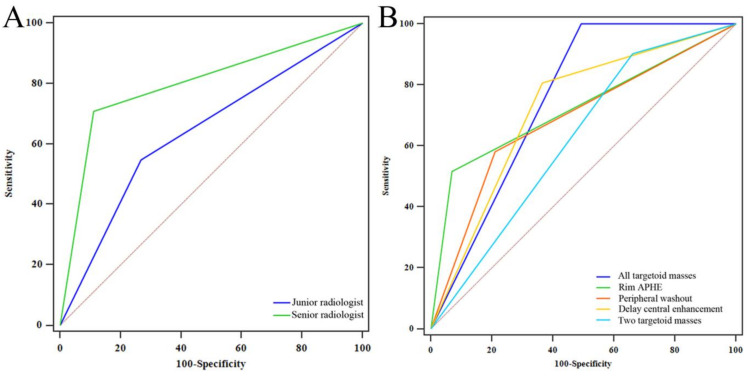
Comparison of the receiver operating characteristic curves of junior and senior radiologists (**A**) and each targetoid mass (**B**).

**Table 1 diagnostics-12-01043-t001:** Baseline Clinical Characteristics of LR-M tumours.

Characteristics	Training Cohort (*n* = 72)		Validation Cohort (*n* = 30)		p
	HCC (*n* = 22)	Non-HCC (*n* = 50)	HCC (*n* = 9)	Non-HCC (*n* = 21)	
Age					0.269
≤50 years	10 (13.8%)	20 (27.8%)	5 (16.7%)	4 (13.3%)	
>50 years	12 (16.7%)	30 (41.7%)	4 (13.3%)	17 (56.7%)	
Sex					0.204
Male	17 (23.6%)	31 (43.1%)	6 (20.0%)	10 (33.3%)	
Female	5 (6.9%)	19 (26.4%)	3 (10.0%)	11 (36.7%)	
Location					0.523
Left	12 (16.7%)	17 (23.6%)	6(20.0%)	6 (20.0%)	
Right	8 (11.1%)	27 (37.5%)	3 (10.0%)	9 (30.0%)	
Junction	2 (2.8%)	6 (8.3%)	0 (0.0%)	6 (20.0%)	
Tumour size (cm)					0.550
≤5 cm	14 (19.5%)	27 (37.5%)	6 (20.0%)	13 (43.3%)	
>5 cm	8 (11.1%)	23 (31.9%)	3 (10.0%)	8 (26.7%)	
Tumour Number					0.867
One	22 (30.6%)	37 (51.4%)	9 (30.0%)	16 (53.3%)	
Multiple	0 (0%)	13 (18.0%)	0 (0.0%)	5 (16.7%)	
Pathological diagnosis					1.000
HCC	22 (30.6%)	-	9 (30.0%)	-	
ICCA	-	38 (52.7%)	-	15 (50.0%)	
CHC	-	1 (1.4%)	-	0 (0%)	
metastases	-	8 (11.1%)	-	6 (20.0%)	
carcinosarcomas	-	3 (4.2%)	-	0 (0%)	
ALT					0.178
≤50 U/L	17 (23.6%)	38 (52.8%)	5 (16.7%)	14 (46.7%)	
>50 U/L	5 (6.9%)	12 (16.7%)	4 (13.3%)	7(23.3%)	
AST					0.115
≤40 U/L	17 (23.6%)	31 (43.1%)	4 (13.3%)	11 (36.7%)	
>40 U/L	5 (6.9%)	19 (26.4%)	5 (16.7%)	10 (33.3%)	
γ-GT					0.209
≤60 U/L	12 (16.7%)	14 (19.5%)	4 (13.3%)	3 (10.0%)	
>60 U/L	10 (13.8%)	36 (50.0%)	5 (16.7%)	18 (60.0%)	
ALP					0.138
≤125 U/L	17 (23.6%)	21 (29.2%)	6 (20.0%)	5 (16.7%)	
>125 U/L	5 (6.9%)	29 (40.3%)	3 (10.0%)	16 (53.3%)	
TBiL					0.095
≤20.4 µmol/L	18 (25.0%)	39 (54.2%)	7 (23.3%)	12 (40.0%)	
>20.4 µmol/L	4 (5.5%)	11 (15.3%)	2 (6.7%)	9 (30.0%)	
DBiL					0.246
≤11 µmol/L	19 (26.4%)	41 (56.9%)	7 (23.3%)	15 (50.0%)	
>11 µmol/L	3 (4.2%)	9 (12.5%)	2 (6.7%)	6 (20.0%)	
IBiL					0.181
≤10.2 µmol/L	18 (25.0%)	39 (54.2%)	7 (23.3%)	13 (43.3%)	
>10.2 µmol/L	4 (5.5%)	11 (15.3%)	2 (6.7%)	8 (26.7%)	
TP					0.504
≤85 g/L	22 (30.5%)	49 (68.1%)	9 (30.0%)	20 (66.7%)	
>85 g/L	0 (0.0%)	1 (1.4%)	0 (0.0%)	1 (3.3%)	
Albumin					0.504
≤55 g/L	22 (30.5%)	48 (66.7%)	8 (26.7%)	21 (70.0%)	
>55 g/L	0 (0.0%)	2 (2.8%)	1 (3.3%)	0 (0.0%)	
AFP					0.411
≤13.2 µg/L	12 (16.7%)	40 (55.7%)	5 (16.7%)	19 (63.3%)	
>13.2 µg/L	10 (13.8%)	10 (13.8%)	4 (13.3%)	2 (6.7%)	
CEA					0.117
≤5.5 ng/mL	19 (26.4%)	33 (45.8%)	8 (26.7%)	18 (60.0%)	
>5.5 ng/mL	3 (4.2%)	17 (23.6%)	1 (3.3%)	3 (10.0%)	
CA19-9					0.613
≤37 U/mL	16 (22.2%)	31 (43.1%)	8 (26.7%)	10 (33.3%)	
>37 U/mL	6 (8.3%)	19 (26.4%)	1 (3.3%)	11 (36.7%)	

HCC: hepatocellular carcinoma; ICCA: intrahepatic cholangiocarcinoma; CHC: combined hepatocellular-cholangiocarcinoma; ALT: alanine aminotransferase; AST: aspartate aminotransferase; γ-GT: γ-glutamyl transferase; ALP: alkaline phosphatase; TBiL: total bilirubin; DBiL: direct bilirubin; IBiL: indirect bilirubin; TP: total protein; AFP: alpha-fetoprotein; CEA: carcinoembryonic antigen; CA19-9: carbohydrate antigen199.

**Table 2 diagnostics-12-01043-t002:** Valuable Features of Each Radiomics Model.

Model	Valuable Features						
	Original			Wavelet			
	Firstorder	Shape	GLCM	Firstorder	GLCM	NGTDM	GLSZM
M1				LLL_Skewness	HHL_Imc1		
				LLH_10Percentile			
M2	Minimum		Idmn	LHL_Mean	LLH_DependenceVariance		HLL_LargeAreaLowGrayLevelEmphasis
				HHL_Kurtosis	LHL_Idn		
M3	Minimum			LLH_Skewness	LHL_Idmn	HLH_Strength	
M4	Minimum		Idmn	LHL_Mean	HHL_MCC		
	Skewness						
M5	Minimum			LLH_Skewness			
				LLH_10Percentile			
				LLL_Skewness			
M6	Minimum			LHL_Mean			
				HHL_Kurtosis			
M7	Minimum	Flatness		LHL_Mean	LHL_Idn		
	Skewness			LLH_RobustMeanAbsoluteDeviation	LHL_Idmn		

M1: model based on T2WI; M2: model based on AP; M3: model based on PVP; M4: model based on T2WI + AP; M5: model based on T2WI + PVP; M6: model based on AP + PVP; M7: model based on T2WI + AP + PVP; GLCM: grey-level cooccurrence matrix; NGTDM: neighbourhood grey-tone difference matrix; GLSZM: grey-level size zone matrix.

**Table 3 diagnostics-12-01043-t003:** Discrimination Performance of the Different Models for LR-M Tumours.

Model	Training Cohort (*n* = 72)			Validation Cohort (*n* = 30)			**Delong**
	AUC (95% CI)	Sensitivity	Specificity	AUC (95% CI)	Sensitivity	Specificity	
M1	0.768 (0.647–0.889)	0.56	0.864	0.759 (0.474–0.944)	0.905	0.667	0.663
M2	0.838 (0.739–0.938)	0.9	0.636	0.836 (0.674–0.998)	0.81	0.778	0.7
M3	0.778 (0.657–0.900)	0.7	0.864	0.762 (0.590–0.934)	0.619	0.889	0.88
M4	0.880 (0.802–0.958)	0.84	0.773	0.836 (0.676–0.996)	0.857	0.667	0.631
M5	0.818 (0.707–0.929)	0.74	0.818	0.783 (0.611–0.955)	0.714	0.889	0.738
M6	0.832 (0.735–0.929)	0.74	0.818	0.815 (0.624–1.000)	1	0.556	0.877
M7	0.884 (0.804–0.963)	0.82	0.864	0.873 (0.728–1.000)	0.952	0.667	0.9

AUC: the area under the curve; CI: confidence interval; M1: model based on T2WI; M2: model based on AP; M3: model based on PVP; M4: model based on T2WI + AP; M5: model based on T2WI + PVP; M6: model based on AP + PVP; M7: model based on T2WI + AP + PVP.

**Table 4 diagnostics-12-01043-t004:** Discrimination Performance of the Radiologists and Targetoid Masses for HCC and non-HCC in LR-M Tumours.

Variable	AUC	95% CI	Sensitivity	Specificity
Junior radiologist	0.64	0.539–0.733	0.548	0.732
Senior radiologist	0.799	0.708–0.871	0.71	0.887
Rim APHE	0.723	0.625–0.807	0.516	0.93
Peripheral washout	0.685	0.585–0.773	0.581	0.789
Delay central enhancement	0.72	0.623–0.805	0.806	0.634
Two targetoid masses	0.621	0.519–0.715	0.903	0.338
All targetoid masses	0.754	0.658–0.833	1	0.507

AUC: the area under the curve; CI: confidence interval; APHE: arterial phase hyperenhancement.

## Data Availability

The data presented in this study are available on request from the corresponding author.

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
