# Peer review of "MRI-Based Radiomics Models to Discriminate Hepatocellular Carcinoma and Non-Hepatocellular Carcinoma in LR-M According to LI-RADS Version 2018"

_diagnostics, 2022, doi:10.3390/diagnostics12051043_

Round 1

Reviewer 1 Report

I read with much interest this original manuscript from Zhang and coll. entitled “MRI-Based Radiomics Models to Discriminate Hepatocellular Carcinoma and non Hepatocellular Carcinoma in LR-M according to LI-RADS version 2018”

The Authors provides new insights concerning the application of radiomica to LI-RADS protocol

The subject of this research is really interesting and covers an expanding field of diagnostic radiology

The results are interesting and adequately discussed, although the Abstract, Introduction and Discussion sections needs some improvements before publication

I have the following requests:

1) Please revise the abstract providing the number of subjects that were involved in the analysis

2) Please refer to the relevant analyses concerning the prognostic role of LI-RADS in HCC setting (10.3390/diagnostics12010160) and its possible implications in curative strategies for HCC treatment such as ablation, resection, or liver transplantation (10.3390/cancers13071671 ; 10.1016/j.aohep.2020.06.007; 10.1111/tri.13983).

3) Please provide a flowchart for patient selection in order to show the total number of patients that were managed in the study timeframe

best regards

Author Response

Response to Reviewer 1 Comments

Point 1:  Please revise the abstract providing the number of subjects that were involved in the analysis.

Response 1: Thank you for your suggestion, we have added the number of tumours involved in the analysis in the Abstract section.

Point 2: Please refer to the relevant analyses concerning the prognostic role of LI-RADS in HCC setting (10.3390/diagnostics12010160) and its possible implications in curative strategies for HCC treatment such as ablation, resection, or liver transplantation (10.3390/cancers13071671 ; 10.1016/j.aohep.2020.06.007; 10.1111/tri.13983).

Response 2: Thank you for your recommendation. We have added the relevant analyses concerning the prognostic role of LI-RADS in HCC and its implications in curative strategies for HCC treatment in the Introduction section of the article, and cited several articles you suggested. Thank you so much.

Point 3: Please provide a flowchart for patient selection in order to show the total number of patients that were managed in the study timeframe.

Response 3: Thank you for your advice. We have added the flowchart for patient selection in the form of figure to the Study Population section of the article.

Reviewer 2 Report

Authors constructed the MRI-based radiomics model to differentiate HCC and non-HCC malignancies, both of which were assigned MRI LR-M. They showed that the discrimination efficacy of the radiomics model was significantly better than that of junior radiologists’ visual assessment. The article is very interesting and useful. Thus, the reviewer thinks that the article is acceptable for this journal, but before accept you should address minor comments described below.

Here are some minor comments:

1. Please show baseline pathological diagnosis and number of the tumors included in the study: HCC, ICC, combined HCC, or liver metastasis.

2. Also, please show histopathological differentiation of HCC included in this study.

3. Did you use background liver tissue information to construct radiomics mode?

If no, please state the reason.

Author Response

Response to Reviewer 2 Comments

Point 1: Please show baseline pathological diagnosis and number of the tumors included in the study: HCC, ICC, combined HCC, or liver metastasis.

Response 1: Thank you for your advice. We have added the baseline pathological diagnosis and the number of various types of tumours in the study in Table 1 and the Tumours Characteristics section.

Point 2: Also, please show histopathological differentiation of HCC included in this study.

Response 2: Thank you for your reminder. We have added the histopathological differentiation information of HCC to the Tumours Characteristics section.

Point 3: Did you use background liver tissue information to construct radiomics mode? If no, please state the reason.

Response3: We are very sorry to tell you that we did not use the background liver tissue information to construct the radiomics models in our study, because we want to use the real tumour information to construct the radiomics models as much as possible, we have added more detailed ROI delineation methods to the image segmentation section of this article. However, since liver tumours may have microvascular invasion, which will affect the prognosis of patients and the choice of treatment strategies, we consider adopting the method of automatically expanding the range of the ROI in subsequent studies, so that the collected datas contain informations of possible microvascular invasion. Thank you very much for your suggestion.

Round 2

Reviewer 1 Report

The Authors significantly improved the paper quality after this round of revision

I feel the paper is suitable for publication

Congratulations for the interesting research